# Characterization of PARP6 Function in Knockout Mice and Patients with Developmental Delay

**DOI:** 10.3390/cells10061289

**Published:** 2021-05-22

**Authors:** Anke Vermehren-Schmaedick, Jeffrey Y. Huang, Madison Levinson, Matthew B. Pomaville, Sarah Reed, Gary A. Bellus, Fred Gilbert, Boris Keren, Delphine Heron, Damien Haye, Christine Janello, Christine Makowski, Katharina Danhauser, Lev M. Fedorov, Tobias B. Haack, Kevin M. Wright, Michael S. Cohen

**Affiliations:** 1Hospital & Specialty Medicine, VA Portland Health Care System, Portland, OR 97239, USA; vermehre@ohsu.edu; 2Translational Oncology Program, Knight Cancer Institute, Oregon Health & Science University, Portland, OR 97210, USA; 3Department of Physiology and Pharmacology, Oregon Health & Science University, Portland, OR 97210, USA; jeffreyhuang21@gmail.com (J.Y.H.); madisonlevinson@arizona.edu (M.L.); 4Vollum Institute, Oregon Health & Science University, Portland, OR 97239, USA; pomavill@ohsu.edu (M.B.P.); wrighke@ohsu.edu (K.M.W.); 5Department of Cell, Developmental, and Cancer Biology, Oregon Health & Science University, Portland, OR 97239, USA; 6Clinical Genetics, Geisinger Medical Center, Danville, PA 17822, USA; sreed4@geisinger.edu (S.R.); gbellus@geisinger.edu (G.A.B.); 7Department of Pediatrics, Brooklyn Hospital, New York, NY 11201, USA; fgilbert515@gmail.com; 8Department of Genetics, Pitié-Salpêtrière Hospital, 75013 Paris, France; bkeren@hotmail.fr (B.K.); delphine.heron@aphp.fr (D.H.); 9Department of Genetics, Hospices Civils de Lyon, 69002 Lyon, France; damien.haye@chu-lyon.fr; 10Department of Pediatrics, Technical University of Munich, 80804 Munich, Germany; Christine.Janello@mri.tum.de (C.J.); Christine.Makowski@mri.tum.de (C.M.); 11Institute of Human Genetics, Technical University of Munich, 81675 Munich, Germany; katharina.danhauser@med.uni-muenchen.de (K.D.); tobias.haack@med.uni-tuebingen.de (T.B.H.); 12Department of Pediatrics, Dr. von Hauner Children’s Hospital, University Hospital, Ludwig Maximilian University of Munich, 80337 Munich, Germany; 13Transgenic Mouse Models Shared Resource, Oregon Health & Science University, Portland, OR 97239, USA; fedorovl@ohsu.edu; 14Institute of Medical Genetics and Applied Genomics, 72076 Tuebingen, Germany

**Keywords:** ADP-ribosylation, MARylation, Parp6, nervous system, dendritic branching

## Abstract

PARP6, a member of a family of enzymes (17 in humans) known as poly-ADP-ribose polymerases (PARPs), is a neuronally enriched PARP. While previous studies from our group show that Parp6 is a regulator of dendrite morphogenesis in rat hippocampal neurons, its function in the nervous system in vivo is poorly understood. Here, we describe the generation of a *Parp6* loss-of-function mouse model for examining the function of Parp6 during neurodevelopment in vivo. Using CRISPR-Cas9 mutagenesis, we generated a mouse line that expressed a Parp6 truncated variant (Parp6^TR^) in place of Parp6^WT^. Unlike Parp6^WT^, Parp6^TR^ is devoid of catalytic activity. Homozygous Parp6^TR^ do not exhibit obvious neuromorphological defects during development, but nevertheless die perinatally. This suggests that Parp6 catalytic activity is important for postnatal survival. We also report PARP6 mutations in six patients with several neurodevelopmental disorders, including microencephaly, intellectual disabilities, and epilepsy. The most severe mutation in *PARP6* (C563R) results in the loss of catalytic activity. Expression of Parp6^C563R^ in hippocampal neurons decreases dendrite morphogenesis. To gain further insight into *PARP6* function in neurons we also performed a BioID proximity labeling experiment in hippocampal neurons and identified several microtubule-binding proteins (e.g., MAP-2) using proteomics. Taken together, our results suggest that PARP6 is an essential microtubule-regulatory gene in mice, and that the loss of PARP6 catalytic activity has detrimental effects on neuronal function in humans.

## 1. Introduction

ADP-ribosylation (ADPRylation) is a reversible regulatory post-translational modification that results in the transfer of ADP-ribose from nicotinamide adenine dinucleotide (NAD^+^) to amino acids on proteins. The enzymes that catalyze ADPRylation belong to a 17-member family known as the poly-(ADP-ribose) polymerases (PARPs), named after PARP1, the first and best characterized member of the PARP family [1,2]. PARPs can be divided into two major subfamilies: 5 PARPs that catalyze the formation of polymers of ADP-ribose (poly-ADP-ribosylation or PARylation) and 11 PARPs that transfer a single unit of ADP-ribose (mono-ADP-ribosylation or MARylation) [3,4,5,6].

*Parp6* is the most abundantly expressed MARylating *Parp* transcript in the developing mouse brain. The protein expression of Parp6 peaks during a critical period of dendritic growth and branching in primary rat hippocampal neurons [7]. Knockdown of Parp6 in embryonic rat hippocampal neurons decreases dendritic complexity in vitro and in vivo [7]. Conversely, overexpression of Parp6 increases dendritic complexity, and this effect depends on the catalytic activity of Parp6. Together, these results show that Parp6 catalytic activity plays a critical role in regulating dendrite morphogenesis during development.

In this study, we sought to further understand the role of Parp6 catalytic activity in the developing nervous system in vivo. To accomplish this, we used CRISPR/Cas9 mutagenesis to generate a Parp6 loss-of-function mouse model. We successfully generated a mouse line that produced a truncated form of Parp6 (Parp6^TR^) that lacked a functional catalytic domain. We found that Parp6 wild-type (Parp6^WT^) embryos and newborns were morphologically indistinguishable from Parp6^TR^ mouse embryos, yet Parp6^TR^ pups turned cyanotic and died within the first hour of birth. Of the few Parp6^TR^ animals that survived, phenotypes included a smaller size, as well as an abnormal locomotor behavior compared to their Parp6^WT^ siblings. During the course of our mouse study, six patients with mutations in PARP6 were identified, showing different degrees of developmental delay, learning disabilities, and epilepsy. Overexpression of a human mutant (PARP6^C563R^) that was catalytically inactive significantly decreased dendritic complexity. Taken together, these results demonstrate that PARP6 is essential in mice and plays a critical role in human neurodevelopment, in part by regulating dendrite morphogenesis.

## 2. Materials and Methods

### 2.1. Animals and Housing

Mouse colony was maintained in standard cages on a daily 12-h light/dark cycle, with food and water ad libitum. Mice were weaned at three weeks of age, ear-tagged, and isoflurane-anesthetized to obtain 2 mm tissue punch biopsies from their ears for genotyping. Males and females were kept as separate groups until old enough to breed (8 weeks old). Embryos (E12.5 to E18) were decapitated prior to removing tail clippings (2–3 mm long) for genotyping.

The use and care of animals used in this study follows the guidelines of the OHSU Institutional Animal Care and Use Committee (Ethics approval code IP00000042, 22.02.21, OHSU IACUC). Female time-pregnant Sprague-Dawley rats were obtained from Charles River Laboratories.

### 2.2. Cell Cultures: Human (HEK293T Cell Line), Rat Primary Cortical and Hippocampal Neurons

Human HEK293T cell cultures, used for their high transfectability, were maintained in Dulbecco’s Modified Eagle’s Medium (supplemented with 10% fetal bovine serum, 1% glutamax and 100 IU penicillin, and 100 μg/mL streptomycin) between 10 and 80% confluency in a 37 °C, 5% CO_2_ tissue culture incubator. Subcultures were made twice a week using 0.05% Trypsin-EDTA.

For primary neuronal cultures, rat cortices (12) and hippocampi (24) were dissected from the brains of E18 Sprague-Dawley rats, and prepared as described in [7], with the following changes: B27 plus Supplement (ThermoFisher Scientific, Waltham, MA, USA) was used instead of GS21, and Neurobasal^TM^ Medium (ThermoFisher Scientific, Waltham, MA, USA) instead of NeuralQ basal medium. Primary cortical neurons were plated at a density of 8 million cells in 10 cm dishes or 250,000 cells per well (24-well plate), while primary hippocampal neurons were plated on coverslips at 250,000 cells per well (24-well plate). All tissue culture dishes, plates, and coverslips were pre-coated with 0.01% poly-D-lysine (Cultrex, R&D systems, Minneapolis, MN, USA). Primary neurons were maintained at 37 °C in a humidified tissue-culture incubator containing 5% CO_2_.

### 2.3. CRISPR: sgRNA Design and Generation of Mouse Line Using Cas9

We used the CRISPR design tool from Feng Zhang’s lab (https://crispr.mit.edu/, accessed on January 2015) to design a single-guide RNA (sgRNA) targeting mouse Parp6 (GenBank accession no. NM_001323522.1). The candidate sequences were checked by BLAST search (https://blast.ncbi.nlm.nih.gov/, accessed on June 2018) to minimize the off-target effects, as well as CRISPR/Cas9 target online predictor (https://crispr.cos.uni-heidelberg.de/, accessed on June 2018) to check for potential off-target genes. To generate the CRISPR/Cas9 vector targeting exon 18 of Parp6, the annealed oligonucleotide corresponding to the sgRNA (sense: 5′-TAGGCATTCAATCCTGCGCAAT-3′, anti-sense: 5′-AAACATTGCGCAGGATTGAATGCC-3′, Figure 1A) was cloned into the *Bsa*I site of pDR274 (#42250, Addgene, Watertown, MA, USA). The T7-PARP6 sgRNA unit was cut out using restriction enzymes, purified, and in vitro transcribed and purified using the mMESSAGE mMACHINE T7 transcription kit (Thermo Fisher Scientific, Waltham, MA, USA). Cas9 mRNA (150 ng/mL) and transcribed sgRNA (100 ng/mL) were mixed and microinjected into the pronuclei of fertilized C57BL/6N eggs, and then transplanted into pseudopregnant C57BL/6N females by the OHSU transgenic mouse resource core (https://www.ohsu.edu/xd/research/research-cores/transgenics, accessed on March 2015). Virgin C57BL/6N females were superovulated to obtain oocytes which were then fertilized to obtain C57BL/6N F1 zygotes. We performed genomic sequencing of mice born to females with transplanted fertilized eggs at postnatal day 21. Of the twelve founder mice obtained (8 females, 4 males), one male had a deletion of five base pairs (TGCGC) plus a single base pair insertion (A), for an overall loss of four base pairs (c.1450_1455delTGTGC, c.1454_1455insA, CCDS 57680.1, Figure 1B). This sequence change resulted in a frameshift mutation 91 residues into the catalytic domain (aa483), and a subsequent early termination codon at amino acid 507 (mouse numbering), which resulted in a truncated PARP6 protein that was devoid of the catalytic domain (Figure 1B).

### 2.4. Genotyping: gDNA Extraction and PCR

For genotyping the animals, we designed forward primers specific to the wild-type Parp6^WT^ or truncated Parp6^TR^ sequence, and reverse primers at different downstream positions giving rise to two distinct PCR products: 743bp for wild-type Parp6 (Parp6^WT^) and 1495bp for a truncated Parp6 (Parp6^TR^), with animals expressing both being heterozygotes (Parp6^HT^) (Figure 1C). Briefly, genotyping of embryonic day E12 and E18 (tail clippings), and postnatal day P21 (ear punches) mice was performed by PCR. Tissues were digested overnight using 20 mg/mL Proteinase K (Roche, Millipore Sigma, St. Louis, MO, USA) in lysis buffer (100 mM Tris-HCl pH 8, 5 mM EDTA, 0.2% SDS, and 200 mM NaCl), 55 °C, with 1000 rpm shaking. The following day, the digested tissue was centrifuged at top speed for 10 min to pellet undigested hair (ear punches, step skipped for tail clippings), and the supernatant transferred to a new Eppendorf tube. gDNA was extracted with phenol followed by phenol:cholorofom (1:1, prepared fresh), and precipitated with 300 mM NH_4_OAc and three volumes of 100% ethanol for at least 2 h at −80 °C. Samples were then centrifuged at 14 Krpm for 30 min at 4 °C, pellet was washed with 75% ethanol, centrifuged at 9.5 Krpm for 5 min at 4 °C, and air-dried. gDNA was resuspended in 30 μL of dH_2_O, and quantified using a Nanodrop One (Thermo Fisher Scientific, Waltham, MA, USA). PCR for Parp6^WT^ and Parp6^TR^ alleles was performed using primer pairs listed in Appendix A. The typical 25 mL PCR reaction mix contained 2.5 U LongAmp DNA Polymerase (New England Biolabs, Ipswich, MA, USA), 0.4 µM forward and reverse primer pairs, 3% DMSO, 300 µM dNTP mix, and genomic DNA template (20 ng). PCR conditions we standardized: 94 °C for 1 min; at 94 °C for 30 s, 58 °C for 1 min, and 65 °C for 75 s for 35 cycles; followed by a final 65 °C 10 min step, using a C1000 Touch Thermal Cycler (BioRad, Hercules, CA, USA). For sexing the embryos, we used murine SRY primers to obtain genotyping primer pairs (http://mgc.wustl.edu/protocols/pcrt, accessed on January 2017), GenBank accession no. NM_011564.1 (Appendix A), and the following PCR conditions: 94 °C for 1 min; 35 cycles of 94 °C for 30 s, 62 °C for 1 min, and 65 °C for 20 s; followed by a final 10 min extension 65 °C, obtaining a 273-bp fragment for animals with a Y chromosome. PCR products were resolved with ethidium bromide-stained 1.8% agarose gel.

### 2.5. Total RNA Extraction from Parp6^WT^ and Parp6^TR^ Brain for cDNA, RT-PCR and RNAseq

Total RNA from *Parp6^WT^* and *Parp6^TR^* E18 embryonic brains was isolated using TRIzol reagent (Thermo Fisher Scientific, Waltham, MA, USA) according to the manufacturer’s protocol. Total RNA, 1 ug, was reverse-transcribed using the ProtoScript II first strand cDNA synthesis kit (New England Biolabs, Ipswich, MA, USA) with random hexamers and oligo dT primers following the manufacturer’s instructions. cDNA was amplified by PCR using LongAmp Taq DNA polymerase (New England Biolabs, Ipswich, MA, USA) in a C1000 Touch Thermal Cycler (BioRad, Hercules, CA, USA). Cycling conditions were: 94 °C for 1 min; 25 cycles of 94 °C for 30 s, 58 °C for 1min, and 65 °C for 12 s; followed by a final 10 min extension 65 °C. All primer sequences can be found in Appendix A. Sizes for the amplified products obtained were 142 bp for each primer set, 1 and 2. Relative expression of Parp6 gene was normalized relative to the housekeeping gene hypoxanthine guanine phosphoribosyl transferase (*HPRT*, 137bp). RNAseq was performed by the OHSU Massively Parallel Sequencing Shared Resource (Illumina sequencing, San Diego, CA, USA) (https://www.ohsu.edu/research-cores/massively-parallel-sequencing-shared-resource-mpssr, accessed on September 2018).

### 2.6. Immunoblots

Cortices were dissected from the brains of E18 Parp6 CRISPR C57BL/6 mice, snap frozen in liquid nitrogen, and stored at −80 °C. Primary cortical neurons were cultured on a PDL-coated 6-well plate (35 mm well−1, Greiner Bio-One), transfected with shLacZ or shPARP6 lentivirus on DIV7, and harvested at DIV10. Frozen neocortical tissue or cultured neurons were lysed in tissue protein lysis buffer (50 mM HEPES, pH 7.4 + 150 mM NaCl + 1 mM MgCl_2_ + 1% triton x-100) with cOmplete^TM^ protease inhibitors (Roche, Millipore Sigma, St. Louis, MO, USA), and Phosphatase Inhibitor cocktails 2 and 3 (Sigma, Millipore Sigma, St. Louis, MO, USA). Tissues were homogenized using a BeadBug stainless steel bead homogenizer (Benchmark Scientific, Sayreville, NJ, USA) at 4000× speed for 20 sec. Lysates were then centrifuged at 10,000 rpm for 10 min at 4 °C and supernatants were transferred to new sample tubes. Samples were prepared and resolved using 10% SDS-PAGE gels and subjected to Western blot analysis. Once transferred onto nitrocellulose, blots were blocked (5% milk in TBST: 100mM Tris base, 150 mM NaCl, and 0.1% Tween-20), and incubated with anti-Parp6 and anti-βactin (Appendix A) for 2 h at RT with shaking. After washing, membranes were incubated with HRP-linked secondary antibodies (Appendix A), washed, and developed using West Pico or West Femto substrates (Pierce, Thermo Fisher Scientific, Waltham, MA, USA). Blots were then imaged and analyzed (densitometry) using a ChemiDoc system (BioRad, Hercules, CA, USA). Molecular mass markers were from BioRad (Cat# 1610373, Hercules, CA, USA).

### 2.7. Whole Mount Immunohistochemistry and Imaging

Tail clippings were taken from ice-anesthetized E18 Parp6 CRISPR C57BL/6 embryos for genotyping, followed by cutting the skin of the embryo along top of the head and chest for better fixative penetration, and fixed in 4% paraformaldehyde (in 100 mM PBS + 4% sucrose) overnight at 4 °C. Fixed diaphragms and rib cages were dissected out and washed three times in 100 mM PBS containing 1% triton (30 min each wash) and then blocked (PBS containing 1% triton + 10% FBS + 0.2% NaN_3_) for 4 h. Tissues were then incubated in anti-β-tubulin primary antibody (Appendix A) diluted in block solution, at 4 °C and shaking (75 rpm) overnight. The next day, tissues were washed three times in 100 mM PBS containing 1% triton + 10% FBS (45 min each wash), followed by two washes in 100 mM PBS + 1% triton (10 min each). Tissues were then incubated in secondary antibody diluted in block solution without NaN_3_, at 4 °C and shaking (75 rpm) overnight. Three final washes in 100 mM PBS + 1% triton-X100 (15 min each) were prior to mounting in CitiFluor CFM1 with anti-fading media (Electron Microscopy Sciences, Hatfield, PA, USA). Tissues were imaged on an inverted confocal scope (LSM780, Carl ZeissInc., White Plains, NY, USA).

### 2.8. Whole Mount E12.5 Embryo Immunolabeling, Clearing, Imaging, and Quantification

Embryonic day 12.5 (E12.5) embryos were collected from timed pregnant female mice, and tail clips were taken for genotyping and identification. Embryos were fixed overnight in 4% paraformaldehyde at 4 °C in 12-well tissue culture plates, washed in phosphate-buffered saline (PBS) 3x for 30 min at room temperature with gentle shaking, then dehydrated with a methanol (MeOH) series (20, 40, 60, 80, and 100%, in H_2_O) for 1 h at each step at room temperature, followed by overnight incubation in fresh 100% MeOH with agitation at 4 °C. Dehydrated embryos were bleached in 5% H_2_O_2_ in MeOH for 6 h at 4 °C with agitation, then rehydrated through a reverse MeOH gradient (100, 80, 60, 40, and 20%, in H_2_O) for 1 h at each step at room temperature, followed by two 30 min washes in PBS/0.2% Triton X-100. Embryos were then moved to 2 mL tubes and incubated in blocking buffer (PBS/5% Normal Donkey Serum/5% DMSO/0.2% Triton X-100) at 37 °C for 24 h, transferred to fresh blocking solution with mouse anti-neurofilament medium chain (NF165) antibody (Appendix A) for 2 days at 37 °C, washed 5 times for 1 h in PBS/0.2% Triton X-100 at room temperature, then overnight in PBS/0.2% Triton X-100 at room temperature, followed by an incubation in fresh blocking solution with Donkey anti-Mouse conjugated AlexaFluor 488 antibody (Appendix A) for 2 days at 37 °C, protected from light. Finally, embryos were washed 5 times for 1 h in PBS/0.2% Triton X-100 at room temperature, then overnight in PBS/0.2% Triton X-100 at 4 °C, dehydrated through a MeOH gradient (20, 40, 60, 80, and 100%, in H_2_O) for 1 h at each step at room temperature, protected from light, and stored in 100% MeOH overnight at room temperature before clearing and imaging. Embryos were cleared by incubation in a solution of 1:2 benzyl alcohol:benzyl benzoate (BABB) until transparent, moved to fresh BABB, and imaged using a Zeiss AxioZoom stereoscope (Carl Zeiss Inc., White Plains, NY, USA).

Embryo images were analyzed and quantified in FIJI (https://imagej.net/Fiji, accessed on 2016–2018). Eye area was measured by tracing the sharp outline of the iris. Paw area was measured by tracing the sharp outline of the developing forepaw pad. Spinal cord thickness was measured perpendicularly across the dorsal side of the embryo, between the 2nd and 3rd dorsal root ganglia (DRG). Paw innervation outgrowth was measured as the distance from the dorsal funiculus of the spinal cord to the furthest extended axon in the forepaw. Abdominal innervation outgrowth was measured at the first DRG under the forelimb, the last DRG before the hindlimb, and at one DRG between the two. Bundle length was measured from the dorsal funiculus, through the corresponding DRG, and out to the longest axons of a somite bundle.

### 2.9. Cloning: Parp6 RNA Interference and Expression Constructs

The shParp6^WT^-expressing vector targeting the coding sequence of *Rattus norvegicus Parp6* mRNA (GenBank accession no. NM_001106828) and the shLacZ-expressing control vector were subcloned as described in [7]. For shRNA sequences, see Appendix A.

The original N-terminus GFP wild-type Parp6 (GFP-Parp6^WT^) expression construct was cloned into the pCAG vector as described in [7], and then transferred to pEGFPC1 (ClonTech, Takara Bio USA, Mountian View, CA, USA) by PCR amplification (primer sequences in Appendix A) and the restriction enzymes *Age*I and *BsaW*I. The extra seventeen Parp6 residues found in neurons (KRHSWFKASGTIKKFRA) were subcloned into the pCMV-GFP-Parp6^WT^ by designing a short DNA sequence (gBlocks, Thermo Fisher Scientific, Waltham, MA, USA, nucleotide sequence in Appendix A) and using *Kpn*I and *BamH*I restriction sites. The Parp6 clinical point mutations (PARP6^C563R^ and PARP6^R483H^) and the truncated Parp6 (equivalent to the CRISPR mutation) were constructed by designing gBlocks (nucleotide and amino acid sequences in Appendix A) containing the partial PARP6 sequence plus the desired mutations, that were then PCR-amplified with primers containing complementary restriction enzyme digest sequences for subcloning into the neuronal GFP-Parp6^WT^ vector (*BamH*I and *Mlu*I).

The Myc-BirA-Parp6 construct was obtained by transferring the Parp6^WT^ sequence into the Myc-BirA* R118G construct (#35700, Addgene, Watertown, MA, USA), using the restriction enzymes *Not*I and *EcoR*I, followed by the transfer of the Myc-BirA or Myc-BirA-Parp6^WT^ units into the LVEG vector modified to include a P2A-GFP sequence downstream of BirA, using the enzymes *Fse*I and *Xba*I. This BirA-Parp6^WT^-LVEG vector was used along with pLP1, pLP2, and pLP-VSVG constructs to make viral particles using HEK293T cells, CalPhos transfection reagents (ClonTech, Takara Bio USA, Mountian View, CA, USA), and PEG6000, for later cortical neuron transduction.

### 2.10. Transduction of Cortical Neurons, BioID, NeutrAvidin Enrichment, and LC-MS/MS Analysis

The labeling of Myc-BirA-Parp6^WT^ interactors with biotin was performed as previously reported [8]. Briefly, primary cortical neuronal cultures (8 million cells/dish) were transduced on day in vitro (DIV) 4 with lentiviral particles expressing either Myc-BirA-P2A-GFP (control) or Myc-BirA-Parp6^WT^-P2A-GFP, and treated on DIV7 with media supplemented with 50 µM biotin for 24 h prior to protein extraction (DIV 8). The biotinylated proteins were subjected to NeutrAvidin agarose enrichment (for detailed LC-MS/MS sample preparation see [9]). MassSpec experiments were performed using an Orbitrap Fusion (ThermoFisher Scientific, Waltham, MA, USA) equipped with a nanospray UPLC system, followed by processing and analysis thresholds (detailed description in [10]) (https://www.ohsu.edu/proteomics-shared-resource, accessed on April 2017).

### 2.11. PARP6 Immunoprecipitation Auto-MARylation Activity Assay

The MARylation activity assay was performed as described in [7]. Briefly, clarified lysates of HEK-293 transfected with the different Parp6 expression constructs were immunoprecipitated with GFP-trap dynabeads (Chromotek, Planegg-Martinsried, Germany) for 1 h at 4 °C with rotation. Beads were washed three times with lysis buffer (50 mM HEPES pH 7.4 + 150 mM NaCl + 1 mM MgCl_2_ + 1% triton x-100) and incubated with alkyne NAD^+^ in the presence of 3 μM veliparib (Selleckchem, Pittsburgh, PA, USA) to inhibit PARylation by PARP1/2 for 2 h at RT, with shaking. Beads were washed two times with PARP reaction buffer (PRB buffer) and once with PBS. Click reaction was performed for 1 h at RT in PBS containing 1 mM CuSO_4_, 100 µM TBTA, 100 µM biotin, and 1 mM TCEP. Proteins were then eluted by boiling the beads for 5 min in 1.5x Laemmli buffer, separated by 10% SDS-PAGE, and transferred to nitrocellulose membranes (Amersham Protran 0.45 NC, Millipore Sigma, St. Louis, MO, USA). Blots were probed with HRP-strepavidin (MARylation) and anti-GFP antibody (normalization), and developed using West Pico or West Femto substrates (Pierce, Thermo Fisher Scientific, Waltham, MA, USA). Blots were then imaged and analyzed (densitometry) using a ChemiDoc system (BioRad, Hercules, CA, USA). List of antibodies used can be found in Appendix A.

### 2.12. Transfection of Rat Hippocampal Cultures, Immunostaining, Imaging, and Sholl Analysis

On DIV7, rat hippocampal neurons, cultured on coverslips in 24-well plates, were co-transfected with two plasmids using Lipofectamine 2000 Transfection Reagent (ThermoFisher Scientific, Waltham, MA, USA): 1 μg total plasmid DNA (0.8 μg GFP-PARP6 construct + 0.2 μg mApple construct as a filler) in 100 μL OPTI-MEM medium (ThermoFisher Scientific, Waltham, MA, USA) was combined with 1.78 μL Lipofectamine 2000 in 100  μL total volume OPTI-MEM medium. Lipofectamine 2000 reaction mixtures were incubated for 30 min at room temperature. One-half of culture media was removed and saved for later use as conditioned culture media. Reaction mixtures were added dropwise to cells and incubated at 37 °C/5%CO_2_ for 4 h. Transfection media was replaced with one-half conditioned culture media and one-half fresh culture media, and cells were returned to the 37 °C/5% CO_2_ incubator until fixing (PBS containing 4% PFA + 4%sucrose) on DIV10 for 20 min at RT.

Fixed hippocampal neurons were washed three times in 100 mM PBS containing 0.1% Tween-20 (5 min each wash) and mounted in ProlongTM Gold anti-fade mountant with DAPI (ThermoFisher Scientific). Fluorescent images of primary neurons were acquired on a Zeiss Apotome widefield microscope (Carl Zeiss, Inc., White Plains, NY, USA) using the ZEN imaging software (Carl Zeiss, Inc., White Plains, NY, USA). For Sholl analysis, neurons were imaged with a 20x objective lens at a single z-section to fully capture the dendritic tree. Dendritic complexity was analyzed at 50 μm increments from the cell soma using FIJI software (https://imagej.net/Fiji, accessed on 2016–2018) with the Concentric Circles plugin (ImageJ).

### 2.13. Whole Exome Sequencing (WES), Sequence Alignment, and Analysis 

For the c.332C>T and c.766C>T mutations, WES was performed in trio (probands + parents) with the SeqCap EZ MedExome Library kit (Roche, Millipore Sigma, St. Louis, MO, USA) and NextSeq 300bp high output sequencing kit (Illumina system, San Diego, CA, USA), followed by alignment with BWA mem 0.717 and variant calling with GATK haplotype caller 3.8. For the c.485R>H and c.485R>C mutations, WES was done by GeneDx, followed by NextGen sequencing (Illumina system, San Diego, CA, USA) with 100 bp or greater pair-end reads, aligned to human genome build GRCh37/UCSC hg 19, and analyzed for sequence variants (Xome Analyzer, Gaithersburg, MD, USA). For the c.583C>R mutation, an Agilent SureSelect XT library preparation kit in combination with the Human All Exon V5 target region (Agilent Technologies, Santa Clara, CA, USA) was used to analyze the coding regions. Prepared libraries were sequenced as 2 × 100 paired-end reads on a HiSeq2500 system (Illumina, San Diego, CA, USA) to an average >120x coverage. At the time of the initial analysis, read alignment was performed with the Burrows–Wheeler Aligner (v.0.5.8). SAMtools (v.0.1.7) was used for detecting single-nucleotide variants and small insertions and deletions. Sequence and copy number variants were reported according to the Human Genome Variation Society (HGVS) and International System for Human Cytogenetic Nomenclature (ISCN), respectively. All subjects gave their informed consent for inclusion before participation in the study. The studies were conducted in accordance with the Declaration of Helsinki, and the protocol was approved by the Ethics Committee (approval number 2341/09, 24.03.2009).

### 2.14. Statistical Analysis

All graphs and statistical comparisons were generated using GraphPad Prism software (version 9, San Diego, CA, USA). Statistical analyses were performed using the two-tailed paired *t*-test or one-way ANOVA followed by Tukey’s HSD test. All data are presented as mean ± SEM.

## 3. Results

### 3.1. Generation of a Parp6 Knockout Mouse Line Using CRISPR-Cas9 Mutagenesis

Parp6 is expressed preferentially in the mouse nervous system (www.informatics.jax.org and www.Biogps.org databases, accessed on 2015–2021). In the mouse brain, Parp6 is expressed almost exclusively in neurons (http://dropviz.org/ and https://amckenz.shinyapps.io/brain_gene_expression/, accessed on 2015–2021). shRNA- and miRNA-based knockdown experiments in rat hippocampal neurons in vitro and in vivo (using in utero electroporation of shRNA-based plasmids) demonstrate that PARP6 regulates dendritic arborization [7]. To further study the role of Parp6 in the nervous system in vivo, we generated a Parp6 knockout mouse line using the CRISPR-Cas9 mutagenesis system, using a single-guide RNA (sgRNA) sequence corresponding to exon 18 in the C-terminal catalytic domain of Parp6 (Figure 1A). The mutation resulted in a frameshift mutation 91 residues into the catalytic domain (aa483), and a subsequent early termination codon at amino acid 507 (mouse numbering), which resulted in a truncated Parp6 protein (Parp6^TR^) that was devoid of the catalytic domain (Figure 1B, Appendix A). RNAseq analysis from mouse hippocampal tissue showed a 3-fold decrease in *Parp6* RNA levels in Parp6^TR^ compared to Parp6^WT^ samples (Appendix A). Importantly, the expression levels of all other known *Parp* transcripts were not affected in the *Parp6* mutant line created (Appendix A).

Once the genotyping protocol was established, the mouse pups were weaned and genotyped. In general, the crossing of heterozygous Parp6^TR^ males and females originated a genetic ratio of 139:239:8 (n = 386; 36.01:61.92:2.07%), with only six mutants surviving longer than 24 h: three were P23 days old when they suddenly died, two were sacrificed at P21 days old, one male lived to be 100 days old and was sterile, and two P0 pups lived less than one hour (see below). This Mendelian ratio differed significantly from the expected 1:2:1 ratio (Chi-square (2, n = 386) *p* < 0.0001), indicating that mice expressing Parp6^TR^ generally do not survive to adulthood.

To rule out that the death of homozygous Parp6^TR^ was due to an off-target effect of the sgRNAs, we used an online CRISPR/Cas9 target online predictor tool to identify potential off-target sequences for our Parp6 sgRNA sequence, then PCR-amplified and sequenced the top five hits for possible off-target genes: Cacna2da, Pstpip1, Rims2, Atb2b2, and Zfp708. No mutations were found in any of these genes (data not shown). In addition, we backcrossed our heterozygote Parp6^HT^ mice to new C57BL/6N Parp6^WT^ mice and obtained the same genetic ratios after following this line for eight generations.

To further validate our Parp6 knockdown mouse line, we performed RT-PCR reactions to look at the reduction in *Parp6* transcript levels in the Parp6^TR^ embryos compared to the Parp6^WT^ embryos. Because the indel was in the catalytic domain, we designed two sets of primers that recognized the 5′ end of the *Parp6* transcript to look at *Parp6* transcript stability and used these specific primer sets on total RNA obtained from whole brains of Parp6^WT^, Parp6^HT^, and Parp6^TR^ animals. The overall reduction in the *Parp6* transcript in Parp6^TR^ relative to Parp6^WT^ animals was 86.2% (n = 4), while the Parp6^HT^ showed almost no reduction compared to Parp6^WT^ (Figure 2A). These results suggest that the *Parp6^TR^* mRNA is not stable and is likely rapidly degraded. Similar results have been obtained in other CRISPR/Cas9 knockout studies [11].

We next evaluated Parp6 protein levels in Parp6^WT^, Parp6^HT^, and Parp6^TR^ mouse brains by Western blotting. We used a Parp6 antibody generated by the Chang lab, which recognizes a region within the C-terminal catalytic domain of Parp6 [3]. We used this antibody because we found that all commercially available Parp6 antibodies were unreliable (data not shown). We first validated this antibody in rat cortical neurons transduced with a validated shRNA-based knockdown lentivirus that targeted Parp6 [7] (Figure 2B). We observed a complete loss in full-length Parp6 protein levels in Parp6^TR^ compared to Parp6^WT^ and Parp6^HT^ brain lysates (Figure 2C). Because we do not have a Parp6 antibody that recognizes the N-terminal region, we cannot determine if Parp6^TR^ is actually expressed in the Parp6^TR^ mice.

### 3.2. Characterization of Parp6^TR^ Embryos and Few Surviving Adults

In our previous studies, we found that Parp6 regulates dendrite morphogenesis in embryonic rat hippocampal neurons [7]; however, we did not examine axon outgrowth. Parp6 is not only expressed in central nervous system (CNS) neurons, but also in peripheral nervous system (PNS) neurons (e.g., dorsal root ganglion and spinal cord cells) (www.Biogps.org, https://gp3.mpg.de and https://mouse.brain-map.org databases, accessed on 2015–2021). To examine peripheral nerve development, we performed whole mount immunolabeling for neurofilament medium chain (NF165) on embryonic day 12.5 (E12.5) Parp6^TR^ and Parp6^HT^ embryos (Appendix A). Immunolabeled and cleared embryos were imaged and overall embryo development was quantified through eye and paw area measurements. Peripheral nerve development was quantified through outgrowth measurements of the forepaw axon bundle and intercostal axon bundle outgrowth. No significant difference in overall development or axonal outgrowth was seen between Parp6^HT^ and Parp6^TR^ animals from multiple litters (Appendix A). We also compared Parp6^WT^ and Parp6^HT^ animals from a single litter and saw no significant difference (*t*-test for same measurements shown in Appendix A: *p* = 0.2423, *p* = 0.3165, *p* = 0.2141, *p* = 0.4639, and *p* = 0.5953, respectively, n = 3–4). These results demonstrate that catalytic activity of Parp6 is not required for axon outgrowth during development in mice. 

Because Parp6^TR^ newborn pups did not survive more than 24 h, we wanted to identify when Parp6^TR^ animals were dying. We genotyped E18 embryos and observed a genetic ratio of 83:114:76 (n = 273; 30.4:41.8:27.8%; Chi-square (2, n = 273) *p* = 0.1445) indicating that the PARP6^TR^ embryos survived at least until E18 based on Mendelian genetic ratios. In addition, based on their gross morphology, Parp6^TR^ E18 embryos were indistinguishable from Parp6^WT^ and Parp6^HT^ (Figure 3A). The weights of E18 embryos obtained from two pregnant females, even though one litter was larger in overall size than the other, showed no significant differences between the three genotypes (Figure 3B; one-way ANOVA *p* = 0.4087 and *p* = 0.1027), and when weights were normalized to Parp6^WT^, Parp6^HT^ and Parp6^TR^ were 104 and 97% of the Parp6^WT^, respectively.

The only two Parp6^TR^ mice that survived at least to 21 days old were significantly smaller than their Parp6^WT^ and Parp6^HT^ siblings (WT 11.818 ± 3.368 g, HT 9.561 ± 1.792 g, TR 3.675 ± 0.035 g) (Figure 3C, one-way ANOVA *p* = 0.0066). When weights were normalized to Parp6^WT^, Parp6^HT^ and Parp6^TR^ were 80 and 31% of the Parp6^WT^, respectively. In addition to the significantly smaller size at P21 days, these animals showed a marked difference in their motor activity (Appendix A). Their behavior was erratic and jumpy, and the twitching behavior was lost over time in the male that lived 100 days, who eventually caught up in size and weight to his Parp6^WT^ and Parp6^HT^ siblings, making him indistinguishable from his siblings. The only clear phenotype in the long-term surviving Parp6^TR^ male was that he was sterile. Interesting to note is that Parp6 is also highly expressed in the testis (www.biogps.org, accessed on 2015–2021), suggesting a possible role for Parp6 in male fertility.

### 3.3. Loss of Full-Length Parp6 Leads to Perinatal Lethality

At birth, the Parp6^TR^ pups were indistinguishable from their siblings, and were able to feed milk from their mothers (Figure 4A). However, in less than 30 min Parp6^TR^ mice slowly became cyanotic and died (Figure 4B). We sought to understand why Parp6^TR^ mice died soon after birth. We hypothesized that the breathing problems in Parp6^TR^ mice could be due to defects in nerve innervation of the diaphragm and/or rib cage muscles at later stages of embryonic development. To test this hypothesis, we looked at nerve innervation of the rib cage muscles as well as the diaphragm in embryos at embryonic day E18 by immunofluorescence. We found that nerve innervation of the rib cage muscles and the diaphragm was also normal in Parp6^TR^ mice (Figure 5A—top and bottom, respectively).

Another possible explanation for the abrupt cessation in breathing is the improper innervation, formation, or connections in the respiratory centers located in the medulla oblongata and pons, which are parts of the brainstem. These centers control the rate and depth of respiratory movements of the diaphragm and other muscles, and the arrhythmic firing of neurons in a few specific brain areas (pre-Bötzinger and Bötzinger) can lead to abnormal breathing and death by asphyxia. We then looked at the expression of *Parp6^WT^* transcripts in different brain regions of 4-week-old mice. Interestingly, the highest expression was seen in the dorsal medulla, which has an important role in processing sensory information from the upper and lower airways for the generation and control of airway protective behaviors (Appendix A). Based on this finding, we asked if there was enough air in the lungs of P0 Parp6^TR^ mice that turned cyanotic and died just after birth. We prepared cross-sections of P0 lungs from Parp6^WT^ and Parp6^TR^ mice and measured the number and area of clear spaces (air). Overall, there were more (389 vs. 288) and larger clear spaces (2403 vs. 1683 µm^2^) in the Parp6^WT^ lungs than in the Parp6^TR^ ones (*t*-test *p* = 0.0029, n = 4, Figure 5B,C). When the clear spaces potentially filled with air are expressed as a percentage of the total area, the Parp6^TR^ showed significantly less air in their lungs compared to Parp6^WT^ (46.08 vs. 24.05%, *t*-test *p* < 0.0001). This could indicate that the breathing control system was not developed correctly or completely.

### 3.4. Human PARP6 Mutations: Clinical Findings

While pursuing studies of Parp6 function in mice, we learned about six patients between the ages of 4 to 28 years old that had mutations in their *PARP6* gene. Of these mutations, four were de novo mutations and the two remaining, a pair of siblings, were congenital with the healthy parents being heterozygous carriers. Whole exome sequencing of the patient DNA samples followed by data analysis, including read alignment and variant calling, identified two mutations in the catalytic domain and two in the non-catalytic domains of the PARP6 protein (Table 1). In general, these patients showed developmental delay, learning disabilities, and epilepsy.

The twenty-eight-year-old male was shown to have a mutation (P111L) in the non-catalytic domain of *PARP6*. He was the second of three siblings born to healthy non-consanguineous parents, following a normal pregnancy and birth weight. He struggled with a persistent food disorder (gastro-esophageal reflux disease), had a laparoscopic Nissen fundoplication at the age of 20 years old, and currently still has problems with solid food. Seizures evolved from the age of three years on, and he is currently on antiepileptic medications. He had delayed motor (started to walk at the age of 4 years, with flexion of knees and prone to falling, showing global and fine motor skills impairment) and language (no words but cries, very limited understanding) development. A micro-array-based comparative genomic hybridization test (array CGH test) was shown to be normal, a screen of an intellectual disability panel of 275 genes was normal, and the FMR1 gene was also normal. The patient does not have hearing or visual problems, and the brain MRI and heart ultrasound were normal.

The four-year-old girl with the second mutation in the non-catalytic domain (H256Y) showed intrauterine growth restriction (IUGR) followed by normal psychomotor development, and partial agenesis of the corpus callosum. The in silico combined annotation-dependent depletion (CADD) score was 25.

With the exception of the P111L mutation, patients with the mutations in the catalytic domain showed more severe symptoms. All four patients with mutations in the catalytic domain showed mild to severe global developmental delays since early infancy, epilepsy, and two of them (R485 mutations) showed speech delay and learning disabilities. Interestingly, the amino acid position 485 was in close proximity to the catalytic tyrosine at position 487 [7]. The de novo R485H mutation was discovered by whole exome sequencing in a 10-year-old boy who was suffering from recurrent focal motor seizures and mild to moderate intellectual disability. Chromosomal microarray analysis was initially performed and was normal. His seizures resolved by the age of 13 years and he no longer requires anti-epileptic medications. He continues to have mild to moderate intellectual disability (ID) and performs approximately two grade levels below his age-related peers. In addition to a history of seizures and ID, he has left renal hypoplasia and hypospadias which was repaired at age 2 years. His height, weight, and head circumference have all been tracking within the normal range on standard pediatric growth charts.

Neurological examination for one of the patients (boy with C563R mutation) revealed overall axial and peripheral hypotonia, globally reduced muscle strength (3/5), and hyperactive deep tendon reflexes (+3). A brain MRI performed at three years of age revealed cortical atrophy and delayed myelination with a paucity of white matter with both supra- and subtentorial involvement, which included the cerebellar hemispheres, vermis, and pons. Other systems were not involved as indicated by the normal abdominal ultrasound, echocardiogram, and the results of general blood hematology and biochemistry. Specifically, he had no unusual infections and no skin sensitivity. Hearing and ophthalmic examination were normal in general. As for his sister with the same C563R mutation, MRI showed macrocephalus with frontotemporal enlargement of the outer cerebrospinal fluid space, and symmetrical hyperintense signal alteration in the dentate nucleus on both sides. Taken together, these clinical phenotypes support a critical role for PARP6 in neurodevelopment in humans.

We next examined the impact of two human mutations (R485H and C563R, both in the catalytic domain) on PARP6 activity and function. We generated GFP-Parp6 expression constructs of these mutants, as well as the Parp6^TR^ mutant (CRISPR Parp6 mutant) (Figure 6A). To examine the catalytic activity of Parp6, we used a GFP-immunoprecipitation (IP)-auto-MARylation assay developed previously in our lab [7]. This assay uses 6-alkyne-NAD^+,^ an NAD^+^ analogue that can be coupled to biotin-azide via the “click” reaction. As expected, GFP-Parp6^WT^ exhibited robust auto-MARylation activity whereas GFP-Parp6^C563R^ was completely inactive, while GFP-Parp6^R485H^ showed a slight reduction in auto-MARylation activity (Figure 6B,C). As expected, GFP-Parp6^TR^ was completely inactive due to the absence of the catalytic domain.

Next, we then examined the effects of two of the human PARP6 mutants on dendritic complexity in primary rat hippocampal neurons. In previous studies we found that overexpression of Parp6^WT^ in primary rat hippocampal neurons increased dendritic complexity, whereas overexpression of a catalytically inactive variant decreased dendritic complexity [7]. Primary neuronal cultures were transfected on DIV7 with GFP-Parp6^WT^ (control), Parp6^C563R^, GFP-Parp6^R485H^, and GFP-Parp6^TR^, together with an mCherry fill, and were fixed on DIV12. We found that overexpression of Parp6^C563R^ and GFP-Parp6^TR^ significantly decreased proximal (50 μm from cell soma) dendritic complexity compared to the GFP-Parp6^WT^ control (Figure 6D). Conversely, overexpression of GFP-Parp6^R485H^ significantly increased distal dendritic complexity (100 μm from cell soma) compared to the Parp6^WT^ control. These overall results show that the Parp6^C563R^ mutant phenocopies the catalytically inactive PARP6 mutants in neurons, consistent with the loss of its catalytic activity. The fact that the Parp6^R485H^ mutant increased dendritic complexity relative to Parp6^WT^ suggests that it acts as a gain-of-function mutant.

The mechanism by which Parp6 regulates dendritic complexity is currently unknown. As a starting point to investigate how Parp6 regulates dendritic complexity, we determined the Parp6 interactome in neurons using BioID proximity labeling [8]. We generated a Parp6 variant in which a promiscuous biotin ligase (Myc-BirA*) was fused to the N-terminus of Parp6^WT^. When cells are treated with media containing exogenous biotin, BirA* will biotinylate nearby proteins in a diffusion-limited process allowing for identification of proximal proteins using biotin affinity capture. We expressed Myc-BirA*-P2A-GFP (control) and Myc-BirA*−Parp6^WT^-P2A-GFP in cortical neurons using lentiviruses, treated with biotin, and observed an increase in biotinylated proteins (Appendix A). To identify potential Parp6 interactors, we captured biotinylated proteins using Neutravidin agarose, performed on-bead trypsin digestion, and subjected resultant peptides to LC-MS/MS analysis (in duplicate). We obtained a list of 89 proteins that were present in either or both runs, which was then reduced to five possible Parp6 interactors after calculating enrichment over control (Appendix A). Interestingly, the top four interactors were microtubule-associated proteins (microtubule-associated proteins 2, 1B, protein tau, and RP/EB family member 2), which play important roles in regulating the microtubule cytoskeleton in neurons, especially during development. These results are consistent with our functional studies and suggest that Parp6 may play a role in regulating the neuronal microtubule cytoskeleton.

## 4. Discussion

In this study we generated a loss-of-function Parp6 mouse line (Parp6^TR^) using CRISPR-Cas9 mutagenesis. Homozygous Parp6^TR^ do not express full-length Parp6, but may possibly express a truncated Parp6 protein that is missing its catalytic domain. Nevertheless, homozygous Parp6^TR^ are devoid of Parp6 catalytic activity. No gross morphological defects were observed during embryonic development. Despite this, 92% of homozygous Parp6^TR^ mice died perinatally. These results support the notion that Parp6 catalytic activity is required for postnatal survival.

Parp6^TR^ newborns appeared to die within the first hour of birth due to suffocation. Post-mortem examination of the lungs from homozygous Parp6^TR^ P0 pups showed fewer air pockets compared to lungs from Parp6^WT^ P0 pups. While we do not know the mechanism of postnatal death in homozygous Parp6^TR^ mice, we speculate that this is due to a loss of Parp6 function in the CNS neurons that regulate breathing. Indeed, we found high levels of *Parp6* transcripts in the dorsal medulla of Parp6^WT^ animals. The dorsal medulla contains the dorsal respiratory group, which is a collection of neurons that are essential for the basic rhythm of breathing. Future studies will focus on examining the function of Parp6 in dorsal respiratory group neurons.

The identification of mutations in *PARP6* in six patients with various neurodevelopmental and motor disorders underscores the importance of *PARP6* function in the nervous system. Unfortunately, the general lack of survival of homozygous Parp6^TR^ mice (missing more than 60% of the catalytic domain) precluded a thorough examination of behavioral and motor functions found in the patients who exhibited single point mutations in *PARP6*. Nevertheless, the only two surviving homozygous Parp6^TR^ mice did exhibit motor defects that were similar to the hypotonia and twitching reflexes observed in some of the young patients with mutations in the *PARP6* gene. Future generations of human Parp6 mutants in mice using CRISPR-Cas9 knock-in strategies will potentially provide an opportunity to examine the postnatal consequences of the loss of Parp6 activity.

While the mechanism by which Parp6 controls dendritic morphogenesis—and likely other neuronal functions—remains unknown, our proximity labeling proteomic studies point to a role for Parp6 in regulating the neuronal microtubule (MT) cytoskeleton. MT-associated proteins (e.g., MAP2) regulating the MT network are essential for many neuronal functions, including dendrite morphogenesis and synapse formation. The major PARP6-interacting proteins we identified were MAP2, MAP1B, EB2, and Tau. MAP1 proteins include MAP1B, which is highly expressed during early neuronal development and involved in shaping the neural tube [12], while MAP2 is enriched within dendrites and was shown to increase MT stability [13]. Tau is expressed in the brain throughout development into adulthood and, due to its implication in neurodegenerative diseases, is the most widely studied MT-binding protein [14]. Mutations in tubulin or proteins that interact with and directly modulate the structure and function of the MT cytoskeleton have led to neurodevelopmental disorders, including intellectual disabilities and autism spectrum disorders [15,16,17]. Given that these clinical phenotypes were observed in the patients with PARP6 mutations, we hypothesize that PARP6 regulates dendrite morphogenesis by modulating the MT cytoskeleton.

Microcephaly, which was observed in the *PARP6^C563R^* patients, is a neurodevelopmental disorder characterized by hypoplasia of the corpus callosum arising from abnormal brain growth, and can be caused by toxic exposures, in utero infections, and metabolic conditions [18]. Microcephaly primary hereditary (MCPH), caused by autosomal recessive mutations, is presented at birth and is relatively rare. Often, MCPH individuals have intellectual disability, language delay, and varying degrees of motor function delay [18,19,20]—clinical features seen in the patients with PARP6 mutations. Microcephaly has also been associated with defects in centrosomes [20,21]. Intriguingly, recent studies show that Parp6 knockdown, or inhibition with a small molecule inhibitor (i.e., AZ0108), in cancer cells induces multipolar spindle formation [22,23]. Given that multipolar spindles are associated with centrosome amplification [24], these results suggest a potential role for Parp6 in maintaining centrosome integrity. Further studies are required to determine if Parp6 regulates the centrosome in neurons, which, interestingly, is the microtubule organizing center [25].

Collectively, our findings provide strong evidence supporting a novel role for PARP6 in neuronal development. Our work lays the groundwork for future studies examining the function of Parp6 in the mouse nervous system and how mutations in *PARP6* impair neuronal function in humans.

## Figures and Tables

**Figure 1 cells-10-01289-f001:**
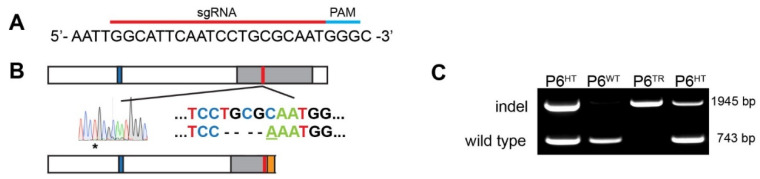
Identification of CRISPR/Cas9-mediated genome modification of Parp6 in mice by sequencing and genotyping analysis. (**A**) sgRNA target site (red) followed by the PAM sequence (blue) in the Parp6 gene. (**B**) Schematic representation of the domain structure of Parp6^WT^: additional sequence found in neuronal cells (blue box), catalytic domain (gray box), sgRNA target site and CRISPR-derived changes (Δ 5bp del + 1bp insert, red box), and frameshift mutation leading to changed residues and an early stop codon (orange box) resulting in a truncated Parp6 (Parp6^TR^). (**C**) Mouse genotyping shows upper 1495bp band representing the indel allele (Parp6^TR^), and lower 743bd band representing the wild-type Parp6^WT^ allele. Parp6^HT^ (heterozygous) shows both alleles.

**Figure 2 cells-10-01289-f002:**
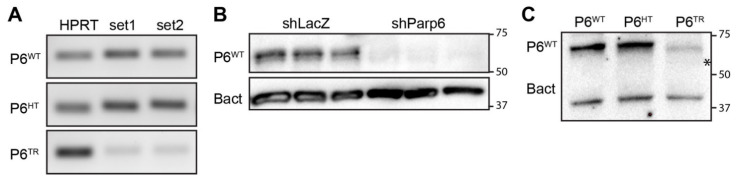
CRISPR-generated Parp6 mutation significantly reduces the levels of full-length *Parp6* transcripts and protein compared to Parp6 wild-type and heterozygous animals. (**A**) RT-PCR image of *Parp6^WT^* and *Parp6^TR^* transcripts levels relative to mouse *HPRT* control, showing a significant reduction in 5′Parp6 transcripts in the CRISPR mutants compared to Parp6^WT^ and Parp6^HT^ (one-way ANOVA, *p* < 0.0001, intensity of bands, n = 4). (**B**) Validation of Parp6 antibody using lysates from cortical neuronal cultures transduced with shParp6 or shLacZ control. Parp6 protein levels relative to β-actin show a significant 97.3% reduction in PARP6 in the shParp6 cortical neurons compared to shLacZ transduced neurons (shLacZ = 0.646 ± 0.067, shPARP6 = 0.017 ± 0.008; *t*-test *p* < 0.0007, n = 3). (**C**) Western blot showing expression of full-length Parp6 relative to β-actin (Parp6^WT^ = 2.037 ± 0.037, Parp6^HT^ = 2.141 ± 0.371, and Parp6^TR^ = 0.320 ± 0.157, mean ± SEM, n = 2) indicating a significant reduction in Parp6 protein in the CRISPR mutants. (*) indicates where the truncated Parp6 protein would be (predicted 57 KDa).

**Figure 3 cells-10-01289-f003:**
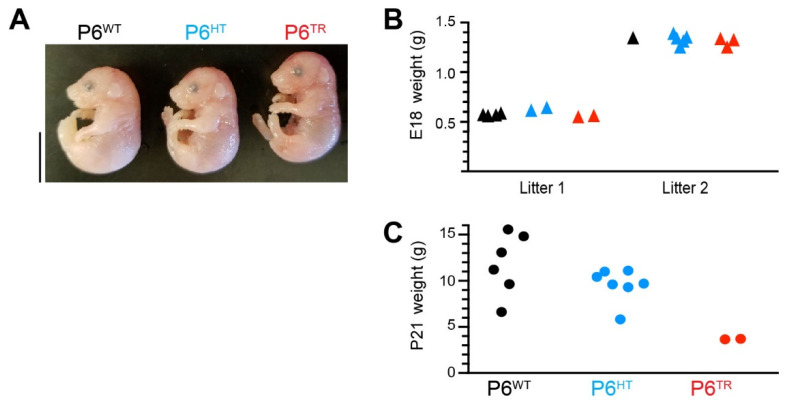
While Parp6^TR^ E18 embryos are indistinguishable from Parp6^WT^ and Parp6^HT^ embryos, P21 Parp6^TR^ mice are significantly smaller compared to Parp6^WT^ and Parp6^HT^ animals. (**A**) Representative image of Parp6^WT^, Parp6^HT^, and Parp6^TR^ embryos age E18, showing that they are indistinguishable from each other at this age. Scale bar: 1 cm. (**B**) Body weights of E18 embryos from two separate litters showing no significant differences (one-way ANOVA Litter 1 *p* = 0.4087 (n = 8), Litter 2 *p* = 0.1027 (n = 9). Combined, Parp6^TR^ and Parp6^HT^ are 97% and 104% of Parp6^WT^, respectively. (**C**) Body weights of P21 animals (one-way ANOVA *p* = 0.0066; Parp6^TR^ and Parp6^HT^ are 31% (*p* < 0.001) and 80% (*p* < 0.05) from Parp6^WT^, respectively).

**Figure 4 cells-10-01289-f004:**
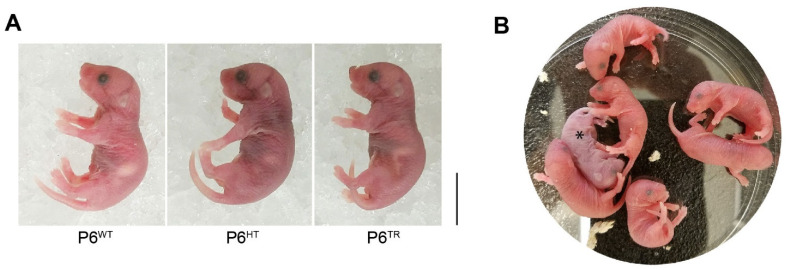
Parp6^TR^ newborns are indistinguishable from Parp6^WT^ animals. (**A**) Representative image of newborns that are under 30 min old, note the stomachs of all three animals with milk. Scale bar: 0.5 cm. (**B**) Representative image of newborns just under 60 min old, note the one animal (Parp6^TR^) that turned gray (*). Scale bar= 0.5cm.

**Figure 5 cells-10-01289-f005:**
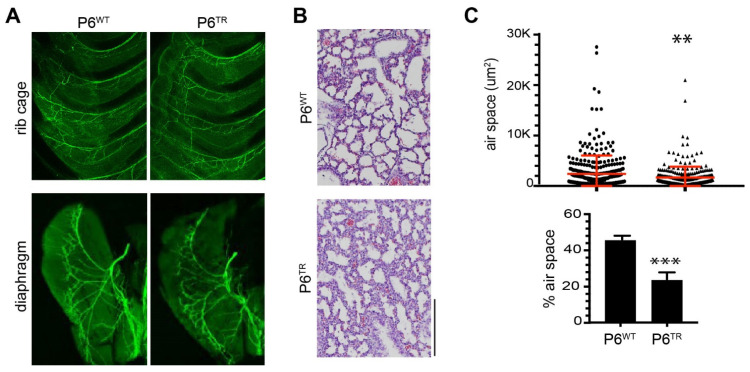
Innervation of rib cage and diaphragm of Parp6^TR^ is indistinguishable from Parp6^WT^ newborns, while Parp6^TR^ animals show less air in their lungs after 60 min of being born. (**A**) Representative images of rib cages (top) and diaphragms (bottom) of E18 Parp6^WT^ and Parp6^TR^ labeled with β-tubulin showing their innervation patterns (top). (**B**) Lung sections of Parp6^WT^ and Parp6^TR^ newborns (under 60 min old) stained with H&E showing size and proportion of open alveolar spaces (air, top). Scale bar = 25 µm. (**C**) Representative distribution of individual alveolar air spaces (µm^2^), showing that individual air spaces are significantly fewer and smaller in Parp6^TR^ compared to Parp6^WT^ (*t*-test *p* = 0.0029, **). (**D**) Percentage of total alveolar space of Parp6^WT^ vs. Parp6^TR^ (*t*-test *p* < 0.0001, ***). n = 389 and 288 for Parp6^WT^ and Parp6^TR^, respectively (bottom). n = 4 images for each genotype.

**Figure 6 cells-10-01289-f006:**
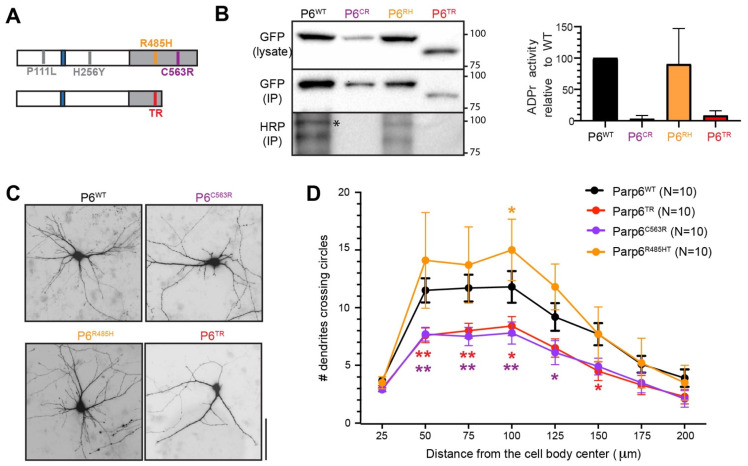
Parp6^TR^ and Parp6^C563R^ expressing hippocampal neurons showed decreased MARylation activity of Parp6, as well as reduced dendritic branching. (**A**) Schematic representation of Parp6^WT^ showing the position of the extra neuronal sequence (blue), the catalytic domain (gray), and the location of the identified clinical point mutations PARP6^P111L^, PARP6^H256Y^, PARP6^R485H/C^, and PARP6^C563R^ found in patients. (**B**) Western blot of the auto mono-MARylation of Parp6^WT^, Parp6^C563R^, Parp6^R483H^, and Parp6^TR^. (*) indicates GFP-Parp6 protein (left), and quantification of ADPr activity of the different Parp6 constructs (right). (**C**) Parp6^R485H^ increased, whereas Parp6^C563R^ and Parp6^TR^ significantly decreased dendritic complexity in primary hippocampal neurons. E18 primary rat hippocampal neurons were co-transfected with mApple (filler) and GFP-Parp6WT (control), GFP-Parp6^C563R^, GFP-Parp6^R485H^, or GFP-Parp6^TR^ on DIV7 and fixed on DIV12. Shown are representative binary images generated from fluorescent images. Scale bar, 20 μm. (**D**) Quantification of results in (**C**) using Sholl analysis. Error bars represent SEM. * *p*  <  0.05, ** *p*  <  0.001 (one-way ANOVA followed by Tukey’s HSD test) compared to Parp6^WT^.

**Table 1 cells-10-01289-t001:** Genetic background of PARP6 mutations. Age is shown in years as (age at diagnosis/current age), zygosity is based on allelic variant frequency, ND = not determined.

Age (y)	28/28	3/5	7/8	10/13	12/17	4/9
Sex	male	female	female	male	male	female
Gene	c.332 C>T	c. 766 C>T	c. 1453 C>T	c. 1454 G>A	c. 1687 T>C	c. 1687 T>C
Protein	p. P111L	p. H256Y	p. R485C	p. R485H	p. C563R	p. C563R
Zygosity status	heterozygous (de novo)	heterozygous (de novo)	heterozygous (de novo)	heterozygous (de novo)	homozygous	homozygous
Other relatives	sporadic	sporadic	Father: no Mother: ND	Sporadic Parents: ND	Parents heterozygous carriers	Parents heterozygous carriers

## Data Availability

Proteomics data is made available via the Proteomics Identification Database (https://www.ebi.ac.uk/pride/, accessed on 2021).

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
