# Peer review of "Characterization of PARP6 Function in Knockout Mice and Patients with Developmental Delay"

_cells, 2021, doi:10.3390/cells10061289_

Round 1

Reviewer 1 Report

The manuscript by Vermehren-Schmaedick et al., describes the role of PARP-6 in neurodevelopment, using elegant in vivo and in vitro models. The findings are important and shed light on PARP-6 cellular role during development.

There are several issues to be addressed:

  1. In the method section, detailed information on humas subjects should be added, including ethical approval and informed consent.
  2. Table 1 should be amended and carefully revised. No “background” for human studies. De novo mutations do not reflect zigosity status.
  3. Please carefully revise gene-protein nomenclature throughout the text according to HUGO guidelines.
  4. microcephaly is often associated with genetic syndromes. Please expand the discussion section.

Minor: in the abstract “it’s function” should be “its function” (line3)

Reviewer 2 Report

The present manuscript is well-structured, well-written and easy to understand.

The author described the generation of a PARP-6 loss-off function mouse model for examining the function of PARP-6 during neurodevelopment in vivo. Using CRISPR-Cas9 mutagenesis, they generated a mouse line that expresses a PARP-6 truncated variant (PARP-6TR) in place of PARP-6WT. The author also learned about six patients between the ages of 4 to 28 years old that have mutations in their PARP-6 gene.

It also addresses a subject that is of great interest in the scientific community. A very large amount of work was involved in the study, and as far as I can determine, the work is solid.  The results are always new or interesting.  

It is very important scientific questions for all the readers and the scientists which involved in nervous system development field. I consider no human results the data also very solid and convincing.

I suggest the author revise the title as below, that is more concise.

Disruption of PARP-6 function leads to postnatal lethality in mice

The author didn’t cite the two very important papers.

The PARP family: insights into functional aspects of poly (ADP‐ribose) polymerase‐1 in cell growth and survival.T. Jubin, A. Kadam, M. Jariwala, S. Bhatt, S. Sutariya, A.R. Gani, S. Gautam, R. Begum. Cell Prolif. 2016 Aug; 49(4): 421–437. 

The PARP Side of the Nucleus: Molecular Actions, Physiological Outcomes, and Clinical Targets.Raga Krishnakumar, W. Lee Kraus.Mol Cell. Author manuscript; available in PMC 2011 Jul 9.

Round 2

Reviewer 1 Report

Major concerns have been addressed. Please use italics for gene names (mutations are found in genes and not in proteins).

Author Response

We thank the reviewer for their positive feedback. We have addressed the minor points raised by the reviewer.